# DGFlow-SLAM: A Novel Dynamic Environment RGB-D SLAM without Prior Semantic Knowledge Based on Grid Segmentation of Scene Flow

**DOI:** 10.3390/biomimetics7040163

**Published:** 2022-10-13

**Authors:** Fei Long, Lei Ding, Jianfeng Li

**Affiliations:** College of Computer Science and Engineering, Jishou University, Jishou 416000, China

**Keywords:** simultaneous localization and mapping, no prior semantic knowledge, dynamic objects, grid segmentation method

## Abstract

Currently, using semantic segmentation networks to distinguish dynamic and static key points has become a mainstream designing method for semantic SLAM systems. However, the semantic SLAM systems must have prior semantic knowledge of relevant dynamic objects, and their processing speed is inversely proportional to the recognition accuracy. To simultaneously enhance the speed and accuracy for recognizing dynamic objects in different environments, a novel SLAM system without prior semantics called DGFlow-SLAM is proposed in this paper. A novel grid segmentation method is used in the system to segment the scene flow, and then an adaptive threshold method is used to roughly detect the dynamic objects. Based on this, a deep mean clustering segmentation method is applied to find potential dynamic targets. Finally, the results of grid segmentation and depth mean clustering segmentation are jointly used to find moving objects accurately, and all the feature points of the moving objects are removed on the premise of retaining the static part of the moving object. The experimental results show that on the dynamic sequence dataset of TUM RGB-D, compared with the DynaSLAM system with the highest accuracy for detecting moderate and violent motion and the DS-SLAM with the highest accuracy for detecting slight motion, DGflow-SLAM obtains similar accuracy results and improves the accuracy by 7.5%. In addition, DGflow-SLAM is 10 times and 1.27 times faster than DynaSLAM and DS-SLAM, respectively.

## 1. Introduction

Currently, visual SLAM has been widely applied in many areas, such as unmanned driving [1,2], AR [3], and intelligent robots [4]. The method of the feature-based visual SLAM systems uses a camera sensor to acquire the image and reconstruct the key points into 3D points based on static assumption. Then, the pose of the camera or robot is solved under strong static constraints, which provides the robot with the ability to map the surrounding environment and the ability to display self-orientation. Currently, some advanced SLAM algorithms have achieved remarkable results, such as ORB-SLAM2 [5] and DSO [6]. However, most of the dynamic objects existing in the real environment are in complex or harsh environments, and the dynamic objects can easily lead to the failure of camera pose estimation because of a lack of reliable static key points. Therefore, reducing the negative impact of dynamic objects on the visual SLAM systems and ensuring the robustness and real-time performance of visual SLAM in complex environments are of great significance for the application of robots in complex dynamic environments.

Currently, the main methods for solving the operation of SLAM systems in dynamic environments are divided into three categories: the pure geometry-based methods [7,8], the pure semantic-information-based methods [9], and the hybrid methods combining semantic information and geometry [10].

The geometric methods have reliable confidence in general cases but are less effective in some special cases. The epipolar geometry method will make the algorithm ineffective when the camera moves along the epipolar direction [11]. Another is the RANSAC method, which requires that most of the key points are static points. When the dynamic points reach a certain number, the RANSAC method cannot obtain the correct result [12].

With the development of deep learning, methods based on semantic information have become the mainstream methods for identifying dynamic objects in SLAM systems [9,10,13]. The semantic segmentation network can find a great quantity of possible moving objects, such as people, cars, and animals, and can give pixel-level masks. However, semantic methods can only provide potential motion information and cannot really identify whether there are moving objects in the found objects. A static book may also be in motion, and a dynamic human body may also be at a stationary moment. According to the paper [10], potentially dynamic objects are those objects that are sometimes in a static state and sometimes in a dynamic state. Moreover, the potential dynamic targets can be simply divided into two categories. One category of potential dynamic targets is considered to be moveable targets based on prior knowledge, which may move now or in the future, such as people, cats, dogs, cars, etc. The other category of potential dynamic targets is considered to be static targets based on prior knowledge, which have the possibility of movement, such as books, computers, chairs, etc. To enhance the performance of recognizing dynamic objects, the current methods combining geometric and semantic methods have been widely used in SLAM systems [10,14,15,16].

In these systems, the geometric method is generally used as a supplementary verification method for the semantic method. Namely, the semantic method is firstly used to find the potential dynamic objects, and the geometric method is then used to accurately find the dynamic objects. This method performs very well in some special scenarios. Currently, the methods combining geometric and semantic information have become mainstream for SLAM systems [10,14,15,16,17]. However, the semantic segmentation networks have two problems: The first problem is that most effective semantic segmentation networks need expensive computation and possess bad real-time performance, while the lightweight semantic segmentation networks possess poor segmentation accuracy, although they have good real-time performance. The second problem is that the semantic segmentation networks cannot detect the dynamic objects without their prior semantic knowledge [15].

Most of the current methods using semantic information perform semantic segmentation on the key frames and compute dynamic or static probability of the key points on the non-key frames and transfer them to the next key frame. However, this method may reduce the accuracy, although it meets the real-time requirements of SLAM systems. In contrast, our proposed method combing k-means clustering segmentation and grid segmentation can simultaneously perform dense segmentation in each frame under the premise of ensuring the real-time operation of SLAM systems. On the other hand, to solve the recognition of moving objects without their prior semantic knowledge, we propose a scene-flow-based method to distinguish dynamic–static regions, which enables visual SLAM systems to work well without relying on prior semantic segmentation networks.

The main contributions of this paper can be summarized as:(1)A novel grid segmentation method is proposed to segment the scene flow, then an adaptive threshold method is designed to roughly detect the dynamic objects. Based on this, a deep mean clustering segmentation method is applied to find potential dynamic targets. This proposed segmentation method can guarantee that each frame can be executed in real time.(2)The results of grid segmentation and deep mean clustering segmentation are jointly used to find the moving objects accurately, and all the feature points of the moving part of the moving objects are removed on the premise of retaining the static part of the moving object.(3)A novel SLAM system called DGFlow-SLAM is proposed by integrating our approach into the RGBD-SLAM system and verified on TUM RGB-D datasets. Experimental results show that the system works well in real time in dynamic environments without relying on prior semantic knowledge.

The remaining parts are organized as follows. Section 2 details the related work. Section 3 introduces our specific theory for solving the SLAM operation in dynamic scenarios. Section 4 presents the analysis of the experimental results. Section 5 is the final conclusion.

## 2. Related Work

Currently, SLAM systems that adopt three solutions in dynamic environments can be divided into two categories, namely the SLAM systems based on camera motion [18,19,20,21,22,23,24,25] and the SLAM systems independent of camera motion [26,27,28]. The former systems need to first calculate the motion of the camera itself, and then judge and distinguish the dynamic target. However, this system fell into the “chicken and egg” problem, so the researchers tried to find the dynamic target before calculating the camera’s own motion.

### 2.1. SLAM Systems Based on Camera’s Own Motion in Dynamic Environment

In this kind of system, the main work can be divided into three categories, which are methods based on improved RANSAC [12], algorithms based on foreground/background model segmentation, and algorithms based on semantic segmentation or target detection.

The first representative algorithm obtains the camera’s own motion through improving the RANSAC [12]. Lu et al. designed an enhanced RANSAC method based on grid division and key point distribution, which assumes that background points are widely distributed throughout the image and the dynamic points are relatively concentrated [18].

The second representative algorithm is based on front/background models. This algorithm requires initial assumption about the environment. Kushwaha et al. proposed a background subtraction technique based on optical flow [19]. Sun et al. proposed an RGB-D-based dynamic environment SLAM method to remove moving objects [20]. In the learning process, the LMedS algorithm [29] and the reprojection error are used to derive the foreground. A foreground model is constructed through the codebook model [30]. Jaimez et al. proposed a joint VO–SF system to find dynamic regions based on energy minimization [21]. On this basis, Scona et al. proposed a StaticFusion system through an energy residual function [22].

The third representative algorithm is to filter the moving objects based on semantic information. This method using object detection or semantic segmentation can easily detect potential moving objects and directly remove these regional feature points. The currently widely used semantic segmentation networks include the highly accurate Mask R-CNN [31], the lightweight SegNet [32], etc. The widely used target detection networks include the very fast YOLO series [33,34], the SSD [35], etc. A representative example is the DS-SLAM proposed by Yu [9]. This system adds a semantic segmentation thread for the SegNet network [32] on the basis of ORB-SLAM2 [5] and directly removes the key points in moving objects. A DynaSLAM system [10] was proposed by Bescos et al., which adds a Mask R-CNN [31] to segment the target object and detect objects that may move or are moving by combining geometric methods. The latest RDS-SLAM system [23] proposed by Liu et al. is based on ORB-SLAM3 [24]. This system adds a semantic thread and a semantic-based optimization thread. On the basis of the former method, RDMO-SLAM [25] adds two threads to, respectively, calculate optical flow and velocity estimation.

### 2.2. SLAM Systems Independent of the Camera Motion in Dynamic Environment

Previous studies tried to find the dynamic regions without relying on camera pose, and the general method is to directly use semantic information or geometric constraints to determine whether the regions are dynamic regions or not. The first representative algorithm is a semantic-based approach. Sheng et al. proposed a Dynamic DSO system [26]. This system first uses Mask R-CNN [31] to find the high dynamic targets and checks whether the four neighborhoods of the pixels in the target area are all static points or not. If so, the target area will be determined to be a static area, and this static background area will be directly placed into the DSO system [6]. The second representative algorithm is the method based on geometric constraints. In [27], Dai et al. used the Delaunay triangulation algorithm [36] to obtain the correlation between the points and determine which target region the points belongs to by comparing the distance changes in the triangle edges in two different frames. The edges connecting in two different property areas will be directly removed, and the area with the largest connected area will be regarded as the static area. In [28], Huang et al. proposed a CluterSLAM based on the prior assumption of motion consistency of the feature points of the same rigid body. First, the motion distance matrix of points on the map is calculated. Next, the sparse motion distance matrix is clustered using the Hierarchical Clustering Algorithm (HAC) [37]. Then, the largest rigid cluster block is regarded as the static background region.

## 3. Design of DGFlow-SLAM

### 3.1. Framework Overview

As shown in Figure 1, our framework is improved based on the RGB-D SLAM of ORB-SLAM2, which has three threads of Tracking, Local Mapping, and Loop Closing. Our main work is the preprocessing process indicated by the red box. In the preprocessing process, we use grid segmentation and k-means clustering segmentation to remove the key points of dynamic objects. Compared with the ORB-SLAM2 system, our framework greatly reduces the negative impact of error points. In addition, we use an acceleration strategy instead of semantic neural networks, so our framework runs faster than other systems. As shown in Figure 1, the system first extracts the ORB feature points at the t-th and t−1-th frames. In the Motion Removal of the preprocessing process, the dynamic feature points are mainly removed by joint k-means clustering segmentation and grid segmentation, and the remaining static feature points are employed to obtain rough poses in the low-cost tracking module. The Mapping thread and Loop Closing thread are used to optimize the rough poses. We can obtain a transformation matrix through the optimized pose. Then, we can use the optimized transformation matrix $T_{t-1,t-2}$ at the previous moment to speed up the preprocessing process at the current moment. This paper adopts the Low-Cost Tracking of DynaSLAM [10]. Compared with ORB-SLAM2, our framework can operate more accurately in dynamic environments on the premise of meeting real-time requirements.

### 3.2. Preprocessing Process

The whole preprocessing process is shown in the red dashed box in Figure 2. In this part, we divide the preprocessing process into two cases: non-initial frame and initial frame. Figure 2 shows the detailed implementation flow of removing the dynamic key point in the non-initial frame state. It can be seen that the process is divided into three modules, namely Segmenting dynamic–static Areas, Moving Object Removal and Low-Cost Tracking. The Segmenting Dynamic–Static Areas module is first executed, and outputs two kinds of dynamic–static segmentation mask images, namely the dynamic–static grid segmentation mask image and the clustering dynamic–static segmentation mask image. Then, the Moving Object Removal module uses the obtained two kinds of dynamic–static mask images to remove the key points in the dynamic area, and the Low-Cost Tracking module calculates the rough camera pose using the static key points. Finally, the Segmenting Dynamic–Static Areas module and the Moving Object Removal module will be executed again to obtain the precise static key points, and the final result will be sent to the Tracking process. This detailed process is: We first compute the rough scene flow of the *t*-th frame by warping the t−1-th frame using the precise transformation matrix $T_{t-1,t-2}\in R^{4}$, where $T_{t-1,t-2}$ means the transformation matrix between the t−2-th frame and the *t*-1-th frame. Then, we can partition clearly the grid mask image into dynamic or static region parts through statistically operating on the scene flow and use the segmented grid mask image to clearly distinguish the dynamic–static attribute of the clustering blocks. Next, both the dynamic–static grid mask image and the deep mean clustering dynamic–static mask image are used to remove the key points of the dynamic objects. A more accurate transformation matrix $T_{t,t-1}$ is calculated in the Low-Cost Tracking process using the remaining static environment key points, and a more accurate scene flow is generated. Now, the obtained precise scene flow can be segmented into foreground and background. Finally, the accurate camera positioning can be obtained using the static matching points in the background.

Unlike the case of non-initial frames, a rough transformation matrix $T_{t_{0},t_{1}}$, where $t_{0}$ means the initial time and $T_{t_{0},t_{1}}$, means the transform matrix from the $t_{1}$-th frame to $t_{0}$-th frame is obtained by the Low-Cost Tracking algorithm in the case of the initial frame, and the rest process in the case of the initial frame is the same as the non-initial frame’s rest process.

In this Figure 2, the preprocessing process, the Segmenting Dynamic–Static Areas module, and the Moving Object Removal module jointly complete the Motion Removal process in Figure 1. The red arrow path means the first Motion Removal process, and the blue arrow path means the Low-Cost Tracking and the second Motion Removal process. When the red arrow process is completed, the blue arrow process is then executed. Note that the result generated by the first Motion Removal process is used as an initial input of the blue arrow process in the Moving Object Removal module. The green arrows indicate the final output of the preprocessing process.The following sections give the details of our approach.

### 3.3. Calculation of Scene Flow

Theoretically, the optical flow of an image contains scene flow and camera self-motion flow, which are caused by the dynamic objects and the camera motion, respectively. Now, we can segment the moving objects using the optical flow with an appropriate threshold. However, the self-motion flow of the camera will interfere with the optical flow of moving objects, which makes it difficult to give the appropriate threshold. So, separating the scene flow from the optical flow will be helpful for the segmentation of moving objects.

FlowFusion [38] proposes a 2D scene flow solution method. Optical flow is the coordinate difference of a same point under a camera coordinate system at different moments. The estimation of the optical flow $\delta x_{t-1\to t}^{of}\in R^{2}$ from time *t* − 1 to time *t* is as follows [38]:(1)$$\delta x_{t-1\to t}^{of}=\pi \left(T_{t}\cdot \left(x_{t-1}+\delta x_{t-1\to t}\right)\right)-\pi \left(T_{t-1}\cdot x_{t-1}\right)$$
where $x_{t-1}\in R^{3}$ represents the coordinate vector of the 3D point in the world coordinate system at time *t* − 1, $\delta x_{t-1\to t}$ represents the difference of the 3D coordinate vector of a certain point between time t−1 and *t*, π(·) represents the mapping from 3D points to 2D points, and $T_{t}\in R^{4}$ represents the camera pose at time *t*. The optical flow consists of the scene flow and the camera self-motion flow. The relationship between 2D optical flow and scene flow can be described as follows [38]:(2)$$\delta x_{t-1\to t}^{sf}=\delta x_{t-1\to t}^{of}-\delta x_{t-1\to t}^{ef}$$
where $\delta x_{t-1\to t}^{sf}$ and $\delta x_{t-1\to t}^{ef}$ represent 2D scene flow and camera self-motion flow, respectively. Camera self-motion flow is an optical flow due to the camera’s own motion. The camera transformation matrix is used to warp the original coordinate values to other camera coordinate systems to obtain new coordinates, and the new coordinates are subtracted from the original coordinates to obtain the camera self-motion flow. Now, the camera self-motion flow is calculated as follows [38]:(3)$$\delta x_{t-1\to t}^{ef}=W\left(x,\xi \right)-x$$
(4)$$W\left(x,\xi \right)=\pi (T\left(\xi \right)\pi^{-1}\left(x,D\left(x\right)\right))$$
where W(·) is a warp calculation and is represented by Equation (Equation 4), which warps the coordinates of a pixel point in one image coordinate system to another image coordinate system to obtain its new coordinates. $x\in R^{2}$ are the pixel coordinates on the image and D(x) represents the depth of point x. T(ξ)∈SE(3) represents the transformation matrix between two different frames, and ξ∈se(3) represents the Lie algebra transformation which contains rotation and translation.

It is very time-consuming to compute the dense optical flow, so this paper uses the PWC-NET neural network [39] to estimate the optical flow. In order to avoid too much error caused by the depth camera during measurement, the threshold value is set to 7 m. The error caused by the depth camera will be less than 2 cm in the range of 4 m, but it will rise rapidly when the range is more than 4 m and can reach almost 4 cm when the range is 7 m. Excessive errors are harmful to our method, so the threshold is set to 7 m. As can be seen from the scene flow image in Figure 2, the static background pixels are almost close to white because the camera’s self-motion flow greatly weakens the static part of the optical flow. In contrast, the foreground pixels are brightly colored because the camera’s self-motion flow does not have much effect on the dynamic part.

In the process of calculating the scene flow, we use the transformation matrix $T_{t-1,t-2}$ from the t−1-th frame to *t*-th frame to replace the roughly calculated transformation matrix between the t−1-th frame and *t*-th frame by the low-cost tracking algorithm. Our approach has two advantages: (1) the iteration speed is accelerated because it reduces one time pose solution process; (2) the information between two sequential frames is effectively correlated.

### 3.4. Algorithm Design for Dynamic Grid Segmentation

It is very time-consuming to directly execute scene flow statistics for each pixel, and the discrete error points will also affect the dynamic–static segmentation of the scene flow. To solve this problem, a grid containing 20 × 20 units is used to divide the scene flow image into N grid regions. The reason for setting the grid region as 20 × 20 is as follows: A large grid will result in too many unnecessary pixels and easily lead to wrongly distinguish the dynamic–static regions. On the contrary, a small grid will consume too much calculation time. In our experiment, we found that the size of 20 × 20 is most suitable for the grid. It can ensure the accuracy of the algorithm in distinguishing the dynamic–static regions without consuming too much calculation time. Then, a 2-norm sum of a grid can be obtained by calculating the total size of the 2-norm of the scene flow of all pixels within a grid, and finally a statistical computing is executed based on the 2-norm sums of all grids. This method can greatly speed up the operation and eliminate the influence of outliers. It is worth noting that the moving objects will cause a larger depth difference than the static area, so we only calculate the depth difference between the pixels within the 2 × 2 range of a grid center and their matching points and incorporate this information into the dynamic–static segmentation calculation.

First of all, since we only perform statistical calculations for the scene in the range of 7 m, all grids which only contain the scenes outside the range of 7 m will be removed. The grids containing the scenes both within and outside the range of 7 m will be determined whether they are valid grids or not, according to Equation (Equation 5). Let the grid be $G_{j}=\left\{p_{i}\in R^{2}|p_{1},p_{2},\ldots ,p_{m}\right\}$, and the valid grid calculation method is as follows:(5)$$\left\{\begin{array}{cc} if\;N\left(G_{j}\left(D_{p_{i}}<7\right)\right)/N\left(G_{j}\right)\geq 0.8, & then\;G_{j}\in GC_{valid}\\ if\;N\left(G_{j}\left(D_{p_{i}}<7\right)\right)/N\left(G_{j}\right)<0.8, & then\;G_{j}\in GC_{invalid} \end{array}\right.$$
where $N(G_{j}\left(D_{p_{i}}<7\right))$ represents the number of pixels with a depth of less than 7 m in the grid $G_{j}$, and $N(G_{j})$ represents the total number of pixels in the grid $G_{j}$. Equation (Equation 5) means that when the ratio of the number of valid pixels to the total number of pixels is less than 80%, the grid $G_{j}$ will be regarded as an invalid grid. Considering that some grids are likely the edge grids, and we need to retain those edge grids to contain enough information, we set the threshold to 0.8. Moreover, if the threshold is too small, the grids containing less information will also be retained, which is not conducive to the segmentation of dynamic regions.

In our method, the grid-based process of dynamic–static segmentation is as follows:**Step** **1:**Remove the invalid grids and keep the valid grids.**Step** **2:**Calculate the 2-norm sum for each grid by calculating the total size of the 2-norm of the scene flow of all pixels within the grid and find the grid with the smallest 2-norm sum among the valid grids.**Step** **3:**The dynamic and the static grids are segmented through the segment model, which is built based on a given threshold value and the grid with the smallest 2-norm sum. The detailed implementation process is given below.

Now, we can calculate the 2-norm size of the scene flow for each pixel as follows:(6)$$d_{i,j}=\sqrt{{\left(d_{x}^{sf}\right)}_{i,j}^{2}+{\left(d_{y}^{sf}\right)}_{i,j}^{2}}$$
where $d_{x}^{sf}$ and $d_{y}^{sf}$ are the elements of the scene flow vector, and *i* and *j* mean the *i*-th pixel and the *j*-th grid, respectively. The scene flow of the *j*-th grid is the sum of the scene flow of the pixels in the *j*-th grid, namely $\sum_{i=1}^{m}d_{i,j}$. We select the grid $G_{min}$ with the smallest scene flow in the valid grids $GC_{valid}$ as the static background grid. Finally, We add the depth difference in the process of grid division statistics.

We derive our depth difference equation according to the principle of the literature [13]. Let $\left(v_{i},u_{i}\right)$ be the coordinate of the upper left corner of the center 2 × 2 area of grid $G_{j}$, and the 2 × 2 depth difference at the center of the grid can be calculated as:(7)$$\begin{array}{c} V_{G_{j}}=\sum_{m=0}^{1}\sum_{n=0}^{1}\left|\begin{array}{c} |ExtD\left(T\left(\xi \right)\pi^{-1}\left(\left(v_{i}+m,u_{i}+n\right),D_{\left(v_{i}+m,u_{i}+n\right)}^{t}\right)\right)\left|-\right|D_{\left(v_{i}^{\prime }+m,u_{i}^{\prime }+n\right)}^{t}| \end{array}\right| \end{array}$$
where ExtD(·) is to extract the depth value of the map point, and $\left(v_{i},u_{i}\right)$ and $\left(v_{i}^{\prime },u_{i}^{\prime }\right)$ are matched by optical flow [40]:(8)$$\left\{\begin{array}{c} v_{i}^{\prime }=v_{i}+d_{x}^{of}\\ u_{i}^{\prime }=u_{i}+d_{y}^{of} \end{array}\right.$$
where $d_{x}^{of}$ and $d_{y}^{of}$ are the elements of the optical flow vector.We use scene flow $S_{G_{j}}$ and grid center depth difference together to find the dynamic grid. Then, we can compute the sum of scene flow and the center depth difference for grid $G_{j}=\left\{p_{i}\in R^{2}|p_{1},p_{2},\ldots ,p_{m}\right\}$ by the following equation:(9)$$S_{G_{j}}=\sum_{l}^{m}d_{i,j}+\gamma V_{G_{j}}$$

The grid with the smallest scene flow value in the valid grid is used as the smallest static background grid $G_{min}$. Now, based on the obtained static background grid $G_{min}$, we can divide all grids into two different parts, namely the dynamic grid part and static grid part by the following equation:(10)$$\left\{\begin{array}{cc} if\;S_{G_{j}}\geq \tau_{1}\sum_{i=1}^{m}d_{i,min}^{G_{min}}+\gamma V_{G_{min}}, & then\;G_{j}\in GC_{dynamic}\\ if\;S_{G_{j}}<\tau_{1}\sum_{i=1}^{m}d_{i,min}^{G_{min}}+\gamma V_{G_{min}}, & then\;G_{j}\in GC_{static} \end{array}\right.$$

Among them, $\tau_{1}$ is 3.3∼3.9, and γ is 10∼20. The reason to set the values of $\tau_{1}$ and γ is that using the transformation matrix $T_{t-1,t-2}$ at the previous moment as the initial transformation matrix $T_{t,t-1}$ to segment the current frame may lead to a high error and affect the gap between $G_{min}$, the two moving scene flows, which makes it difficult to judge the distinction between dynamic–static regions. In this case, when the threshold is set too small, the static background will be wrongly segmented into dynamic regions. Furthermore, if there is no valid grid, namely $\forall G_{j}\in GC_{invalid}$, a threshold must be set to distinguish dynamic–static grids:(11)$$\left\{\begin{array}{cc} if\;S_{G_{j}}\geq \tau_{2}, & then\;G_{j}\in GC_{dynamic}\\ if\;S_{G_{j}}<\tau_{2}, & then\;G_{j}\in GC_{static} \end{array}\right.$$
where $\tau_{2}$ is set to 1500∼2000. Considering a special case that most of the pixels in the picture are moving, to retain some moving points and ensure that there are enough feature points to help SLAM run stably, the threshold is chosen in the range between 1500 and 2000. The reason for setting the threshold to 1500∼2000 is as follows: The scene flow of one pixel point of a vigorously moving object is at least 3, so the value of the scene flows of one grid is at least 3∗(20∗20)=1200. Moreover, the inaccurate transformation matrix will make the error range of the scene flows of one grid be about 0.4∼0.8. Then, the minimum error value of the scene flows of one grid is about 0.4∗(20∗20)+1200=1360, and the maximum error value of the scene flows of one grid is about 0.8∗(20∗20)+1200=1520. Furthermore, the influence of depth difference needs to be considered, and the depth difference between two consecutive frames is about 4∼6cm. According to the value range of γ is 10∼20, the minimum depth difference threshold is 4∗(2∗2)∗10=160, and the maximum depth difference threshold is 6∗(2∗2)∗20=480. Then, we have the threshold range (1360+160≈1500,1520+480=2000). The method to judge whether a grid is a dynamic or static grid is shown in Algorithm 1.
**Algorithm 1** Dynamic–static grid discrimination.**Input:** scene flow image and depth image**Output:** Dynamic–static grid sets $Array_{GC_{dynamic}}$ and $Array_{GC_{static}}$  1:  $Array_{GC_{dynamic}}$,$Array_{GC_{static}}$,$Array_{GC_{valid}}$,$Array_{GC_{invalid}}\leftarrow 0$  2:  $F_{min}=+\infty ,threshold=+\infty$  3:  **for**
$each\;G_{j}$
**do**  4:     $c_{1}=N\left(G_{j}\left(D_{p_{i}}<7\right)\right)/N\left(G_{j}\right)$  5:     **if** $c_{1}\geq 0.8$ **then**  6:        $Array_{GC_{valid}}\leftarrow G_{j}$  7:        **if** $F_{min}>\sum_{i=1}^{m}d_{i,j}$ **then**  8:          $F_{min}=\sum_{i=1}^{m}d_{i,j},G_{min}=G_{j}$  9:        **end if**10:     **else**11:        $Array_{GC_{invalid}}\leftarrow G_{j}$12:     **end if**13:  **end for**14:  **if**
$Array_{GC_{valid}}\neq \emptyset$
**then**15:     $threshold=\tau_{1}*F_{min}+\gamma V_{G_{min}}$16: **else**17:     $threshold=\tau_{2}$18:  **end if**19:  **for**
$each\;G_{j}$
**do**20:     $S_{G_{J}}=\sum_{l}^{m}d_{i,j}+\gamma V_{G_{j}}$21:     **if** $S_{G_{J}}\geq threshold$ **then**22:        $Array_{GC_{dynamic}}\leftarrow G_{j}$23:     **else**24:        $Array_{GC_{static}}\leftarrow G_{j}$25:     **end if**26:  **end for**

### 3.5. Dynamic Segmentation Algorithm Based on Deep K-Means Clustering

The simple grid segmentation method cannot segment all key points of the dynamic target region, and there are some potential dynamic points that need to be removed together. For example, people may move the chair they are sitting on when they get up. Then, the moving chair is a potential dynamic object [9,10,13]. According to the suggestion proposed in the paper of Joint VO–SF [21], we selected 24 classes to segment the image. The number of k-means clusters is set according to the image size, for example, if the image size is 640∗480, then the number of k-means clusters is [640/10]∗[480/10]=6∗4=24. This is given empirically by the Joint VO–SF authors, mainly to provide medium-sized clustering blocks.

Next, the segmented grid mask image is used to make a distinction between dynamic–static attributes of the cluster blocks. Let the dynamic overlapping area between a cluster block image and a segmented grid mask image be *S*, and the entire cluster block is considered to be dynamic if the ratio of *S* to the cluster block is great than $\tau_{3}$. The corresponding method can be shown as:(12)$$\left\{\begin{array}{cc} if\;N\left(GC_{dynamic}\cap B_{i}\right)/N\left(B_{i}\right)\geq \tau_{3}, & then\;B_{i}\in BC_{dynamic}\\ if\;N\left(GC_{dynamic}\cap B_{i}\right)/N\left(B_{i}\right)<\tau_{3}, & then\;B_{i}\in BC_{static} \end{array}\right.$$
where N(Z) represents the number of elements in the set *Z* and $B_{i}$ represents the pixel point set of the *i*-th mean cluster. We found that similar regions are in the same motion, so we set $\tau_{3}$ to be 0.3∼0.5. Finally, we eliminate the potential key points in the dynamic clustering regions. The above implementation process can be shown as Algorithm 2. We did not set up 24 clustering blocks for all experiments but only for the experiments with 640∗480 images. Only 24 clustering blocks are available for 640∗480 image sizes. If the image size is 640∗480, then the number of k-means clusters is [640/10]∗[480/10]=6∗4=24. So, the 24 clusters are not constant numbers in Algorithm 2. When working with larger images or smaller images, one has to go for modifying the number of clustering blocks. Bigger images have more clustering blocks and vice versa.
**Algorithm 2** Dynamic–static K–means clusters discrimination.**Input:** Grid mask image and depth image**Output:** Dynamic–static cluster block sets $Array_{BC_{dynamic}}$ and $Array_{BC_{static}}\leftarrow 0$       K-means clustering was carried out on the depth image, and the clustering was       24 blocks 2: **for**
$each\;B_{j}$
**do**            $c=N\left(GC_{dynamic}\cap B_{i}\right)/N\left(B_{i}\right)$ 4:     **if** $c\geq \tau_{3}$ **then**               $Array_{BC_{dynamic}}\leftarrow B_{i}$ 6:     **else**               $Array_{BC_{static}}\leftarrow B_{i}$ 8:     **end if**     **end for**

### 3.6. Acceleration Strategy

As shown in Figure 3, we use a two-layer pyramid for k-means clustering acceleration. First, we use a 4 × 4 Gaussian kernel to reduce the image to 1/4 of the original image. Then, we perform k-means clustering on the upper image, and the iterative times are no more than 10. The upper image clustering results in 24 clustering blocks, and we map the centroids of these clustering blocks back into the original image. As shown in Figure 3, a brown centroid of one cluster on the upper image is mapped to a blue pixel block with the size of 4 × 4 in the original image. Within the range of blue pixel blocks, the coordinate closest to the average depth is the center of the cluster block. We perform the following calculation for each pixel point $\left(v_{i},u_{i}\right)$ in the pixel block:(13)$$CenterError={∥D_{\left(v_{i},u_{i}\right)}-\frac{\sum_{j=0}^{4}D_{\left(v_{j},u_{j}\right)}}{4}\|}_{2}$$
where $D_{\left(v_{i},u_{i}\right)}$ denotes the depth of pixel point $\left(v_{i},u_{i}\right)$. The pixel point $\left(v_{i},u_{i}\right)$ with the smallest CenterError is used as the mean clustering center of the original image, and then the original pixel points are assigned to each clustering center.

The running time of our method is mainly consumed in the process of k-means clustering iteration. The time complexity of k-means clustering is O(NKt). Where *N* is the number of data objects, *K* is the number of clustering blocks, and *t* is the number of iteration times. In layer 1, *N*, *K*, and *t* are 160∗120, 24, and 10, respectively. In layer 0, *N*, *K*, and *t* are 640∗480, 24, and 1, respectively.

### 3.7. Joint Removal of Dynamic Feature Points

Furthermore, to ensure the small objects can be successfully eliminated, DGFlow-SLAM is proposed by combing the segmenting grid mask image and the segmenting mean clustering mask image. DGFlow-SLAM means that both Algorithms 1 and 2 are first used to obtain the corresponding segmenting mask images, respectively, and then the two different segmenting mask images are superimposed to eliminate the dynamic key points.

## 4. Experiment Results

### 4.1. Experimental Setup

The DGFlow-SLAM proposed in this paper is verified on the public dataset TUM, and the following experiments are carried out on the s/static, w/static w/half, w/rpy, and w/xyz datasets. The s/static dataset contains only slight movement caused by the head or partial limbs of a human body. The w/static dataset contains moderate-intensity movement caused by slowly getting up or the walking of human beings, and we can find the camera moves slightly at the same time by observing this dataset. The rest of the datasets, such as the w/half, w/rpy, and w/xyz datasets, contain violent movement caused by human beings, and we can find these cameras move violently at the same time by observing the corresponding datasets. Moreover, these cameras move along a semicircle on the w/half dataset, rotate along the main axis on the w/rpy dataset, and move along the three axes of xyz on the w/xyz dataset, respectively. Finally, the following experiments are completed using Intel Core i7 and 8G memory.

We employ absolute trajectory error (ATE) and relative pose error (RPE) to conduct quantitative evaluation [41]. Among them, ATE represents the global consistency of errors in the metric system and the overall performance of the system, and RPE represents translational and rotational drift errors and reflects the drift of the system. Finally, the system evaluation is divided into two parts: accuracy evaluation and time evaluation.We used the following enhancement values [10] to measure the effect of our system on ORB-SLAM2:(14)$$\eta =\frac{o-r}{o}\times 100\%$$
where *o* represents the size of the error value of ORB-SLAM2 [5], and *r* represents the size of the error value generated by our method. This enhancement value indicates how much we improved the ORB-SLAM2 improvements. The enhancement values are shown in the last two columns of Table 1, Table 2 and Table 3. On the other hand, we use ablation experiments to validate the performance of our grid segmentation, k-means clustering segmentation, and the combination of both methods. Finally, we analyze the operational effects on a public dataset, as well as on a real dataset.

### 4.2. Accuracy and Runtime Evaluation

#### 4.2.1. Accuracy Evaluation

As shown in Table 1, Table 2 and Table 3, we quantitatively analyzed the ATE and RPE of the four systems. RMSE represents the deviation between the observed value and the true value and reflects the robustness of the system. The standard deviation S.D. represents the degree of deviation and reflects the stability of the system. Table 1, Table 2 and Table 3 show the ATE, the RPE translation, and the RPE rotation, respectively. Compared with the classical ORB-SLAM2 [5], the DGFlow-SLAM decreases the RMSE of ATE, RPE translation, and RPE rotation by 95.66%, 90.56%, and 91.54%, respectively, when they detect the fast motion sequence on the dataset w/xyz, and decreases the RMSE of ATE, RPE translation, and RPE rotation by 26.25%, 9.52%, and 1.14%, respectively, when they detect the static sequence on the dataset s/static. Compared with the rest of the SLAM systems, the DGFlow-SLAM achieves the best results on the dataset s/static. This result shows that the DGFlow-SLAM is more suitable to detect the slow motion objects than the rest of the SLAM systems. In addition to this, compared with the methods with prior semantic knowledge, such as the DS-SLAM [9] and the DynaSLAM [10], the DGFlow-SLAM performances slightly worse on the datasets w/static and w/xyz, but the errors of the three SLAM systems are very close to each other.

We compare the algorithm performance of ORB-SLAM2 [5], DS-SLAM [9], DynaSLAM [10], and DGFlow-SLAM through their estimated camera trajectories. Among them, ORB-SLAM2 is a classic SLAM system without prior semantic knowledge, and DS-SLAM [9] and DynaSLAM [10] are the classic semantic SLAM systems. The data of DS-SLAM [9] and DynaSLAM [10] are derived from RDS-SLAM [23]. In Figure 4, black lines, blue lines, and red lines represent true trajectories, estimated trajectories, and difference between estimated and true values, respectively.

Figure 4 displays the comparison results. We know that ORB-SLAM2 [5] performs poorly on all data sets. Compared with ORB-SLAM2 [5], DS-SLAM [9] shows a great enhancement on the rest of the datasets, except the w/rpy dataset. Compared with ORB-SLAM2 [5] and DS-SLAM [9], DynaSLAM [10] shows a better performance on all the datasets. However, DynaSLAM discards much pose data along the camera’s estimated trajectories when it operates on the w/rpy and w/static datasets. So, the DynaSLAM [10] is not good for the loop detection and dense mapping of camera pose. The estimated trajectories by the DGFlow-SLAM are generally consistent with the true trajectories, and most points are close to the true trajectories on the w/half, w/static, and w/xyz datasets. However, the performance of the DGFlow-SLAM is much worse on the w/rpy dataset. Furthermore, we can see that for the most estimated camera poses, DGFlow-SLAM has fewer errors than DS-SLAM [9] and DynaSLAM [10], and for a few estimated camera poses, DGFlow-SLAM has many more errors than DS-SLAM [9] and DynaSLAM [10]. Compared with DynaSLAM [10], DGFlow-SLAM preserves all the camera poses. Compared with ORB-SLAM2 [5], DGFlow-SLAM greatly enhances the estimated camera poses.

It is worth noting that to improve the accuracy, some static parts of a dynamic object are preserved, which is not conducive to dense map construction. On the other hand, the optical flow method is sensitive to light changes, and the current framework cannot be applied to outdoor areas.

#### 4.2.2. Runtime Evaluation

The real-time performance of a SLAM system is an important indicator. DynaSLAM [10] has the best detection accuracy on almost all dynamic datasets compared with the other SLAM systems. However, DynaSLAM [10] takes 500 ms to complete the detection per frame and cannot meet the real-time requirements of SLAM. DGFlow-SLAM using the real-time PWC-NET network [39] module for optical flow estimation is much faster than DynaSLAM [10] using the Mask-RCNN network [31] module for optical flow estimation. Table 4 shows that the total time of the three modules of the DGFlow-SLAM is about 51 ms, less than the working time of DS-SLAM (65 ms) [9] and far less than t DynaSLAM (500 ms) [10]. The reason for the fast speed is that we use an acceleration strategy. The total time for 10 iterations of k-means clustering runs on 1/4 image is 4 ms, and one k-means clustering calculation at 640∗480 is 3 ms. Therefore, the total time for k-means clustering is 7 ms, which is more than 50% of the total time of the preprocessing process. Compared with the consumption time of 37.57 ms for the Seg-Net network [32] used in DS-SLAM [9] and 195 ms for the Mask-RCNN network [31] used in DynaSLAM [10], our algorithm is much faster. So, DGFlow-SLAM meets the real-time requirements of SLAM.

In summary, our scheme outperforms the RGB-D mode in the original ORB-SLAM2 [5] and achieves similar accuracy as DS-SLAM [9] and DynaSLAM [10]. Compared with DynaSLAM [10], on the dataset w/xyz, DGFlow-SLAM performances slightly worse, but its detection speed is greatly enhanced. In Table 1, the enhancement value of DGFlow-sSLAM is less than 1.8% of the enhancement value of 97.46% of DynaSLAM [10]; however, the detection speed of DGFlow-SLAM is 10 times faster than DynaSLAM [10]. Unlike DS-SLAM [9] and DynaSLAM [10], we do not use semantic segmentation networks, and our proposed method is much faster than semantic segmentation. Our method performs optimally in the case of very slight movements.

### 4.3. Ablation Experiment

We use grid segmentation mask image, k-means clustering segmentation mask image, and their joint segmentation mask image to remove dynamic objects and compare their effects on the dataset w/xyz, respectively. As shown in Figure 5, the grid segmentation mask image is less effective in removing dynamic objects, mainly because it does not remove many potential static parts. The joint image is the best in removing dynamic objects, which is because the grid segmented mask image has some accuracy improvement by removing the small object moving parts. The data in Figure 5 and Table 5 are the median values selected by running DGFlow-SLAM three times.

Table 5 shows the data of the RMSE of the ATE for the ablation experiment. The joint mask image works the best, and the RMSE value is only 0.0234.

### 4.4. Run on a Public Dataset

Figure 6 shows our performance on four TUM datasets, where red and blue points represent dynamic key points and static key points, respectively. From the comparison between scene flow image and optical flow image, we can see that the optical flow is greatly weakened, and the picture in the first column of the second row shows the most obvious change. The whole image is very blurry and the boundary of objects cannot be seen clearly, and the reason is that the camera’s self-motion flow covers the whole image. So, we can see all objects clearly when we remove the camera’s self-motion flow. From the picture in the first column of the last row, we can see that most of two human bodies can be found. However, there are still some areas that are not considered to be dynamic masks, such as the shin of the left person and the leg of the right person when they are static. The method regarding the unmoved human body region as static region will help to solve the camera pose. As shown in the last row of Figure 6, most of the feature points of the dynamic region can be found.

Figure 7 shows the detection of small object motion. In Figure 7a, only one head appears in the picture, and the head is well identified and segmented in Figure 7b using the Algorithm 1. In Figure 7c, the white area indicates that the object is completely segmented using Algorithm 2. Figure 7d shows the corresponding dynamic–static key points obtained by the DGFlow-SLAM. Another example is the ball rolling experiment. Figure 7e is the original image. Figure 7f shows the result of Algorithm 1, and the white area represents the rolling ball. Figure 7g shows the result of Algorithm 2. It can be seen that because the moving objects are too small, the k-means clustering method cannot segment the dynamic targets. Figure 7h shows the dynamic and static key points obtained by DGFlow-SLAM. The above experiments show that Algorithm 1 or Algorithm 2 may fail to segment the small objects in different environments, while the DGFlow-SLAM system can segment the small objects in different environments successfully.

Figure 8 shows that we can detect the slow moving objects by combining the grid and the adaptive threshold because most of the slight movements exceed more than 1 pixel distance, and the adaptive threshold is far less than 1. We define the movement change between 0.8∼2 of scene flow of one pixel in a frame as slight motion. The movement change between Figure 8a,b is very slight, and we can see that the foot of the right human body in the figure has very slight movement. We can see that there is a distinct red area in the lower right of Figure 8c, and this distinct red area is clearly found in Figure 8d,e. We can see that the corresponding dynamic key points are marked with red in Figure 8f, and we will remove the relevant dynamic key points in the system. The above experiments demonstrate that our method is highly sensitive to slight movements.

In Figure 8a, most of the area on the right side of the person is in a static state, and only the feet have very slight movements, which is difficult to see from the figure. However, we are able to accurately segment the slightly moving parts based on the scene flow. We also need to segment and remove the ground part for two main reasons: the first reason is that the reflection of the human body affects the ground optical flow estimation; the second reason is that the optical flow estimation is prone to errors on the ground where the overall luminosity is almost the same. This illustrates the difficulty of applying our method to outdoor areas with strong photometric variations, as it is very sensitive to changes in optical flow. In addition, it is worth noting that a stationary chair will move when the left person stands up, which is a typical potential moving object that can be detected by our method.

### 4.5. Run in Real Environment

To verify that our system can work properly in the real world, we conducted extensive experiments for DGFlow-SLAM in a laboratory environment. In these experiments, we used a RealSense D455 camera to capture a large number of 640 × 480-size RGB images and depth images by hand-held shooting. Figure 9 shows the results of our experiments. The first row is the case of a single person walking. From left to right are the captured RGB image, the calculated 2D scene flow, the grid segmentation mask, the k-means clustering mask, and the dynamic–static key point classification results, respectively. Almost all of the dynamic key points of the human body are correctly recognized and marked in red. The second row is a scene of two people moving towards each other, and almost all of the dynamic key points of the human body are also clearly marked.

## 5. Conclusions

In order to improve the adaptability of classical ORB-SLAM2 [5] in complex and unknown environments, we proposed a new framework DGFlow-SLAM to reduce the impact of dynamic feature points. In this framework, grid segmentation and k-means clustering segmentation are combined to remove dynamic regions, which greatly improves the segmentation performance of dynamic–static feature points. In addition, in order to speed up the iteration and strengthen the connection between two sequential frames, the framework uses the previous solution as the current initial solution to roughly segment the dynamic–static regions. Experiments show that DGFlow-SLAM can clearly detect violent motion and slight motion without prior semantic knowledge. Moreover, DGFlow-SLAM has the best accuracy in detecting slight motion than the other SLAM systems because the threshold of DGFlow-SLAM is determined by the minimum grid scene flow and the corresponding grid depth difference. Finally, compared with the classical semantic SLAM systems, namely DS-SLAM [9] and DynaSLAM [10], DGFlow-SLAM is the fastest, and compared with the classic ORB-SLAM2 [5], DGFlow-SLAM improved the accuracy by 95.58%, 95.66%, and 26.25% on the moderate motion dataset, violent motion dataset, and slight motion dataset, respectively.

However, there is still some work to be conducted for DGFlow-SLAM. For example, our method cannot completely realize dense map construction without dynamic objects. In the future, we will use data association to remove temporarily stationary moving objects from the map. In addition, DGFlow-SLAM can only be applied to indoor scenes at present, and the conversion from indoor scenes to outdoor scenes requires more work. For the semantic SLAMs, their development direction should be to accelerate the speed of using semantic segmentation networks and increase the accuracy of segmentation edges. In addition, SLAMs with no prior semantic knowledge and SLAMs with prior knowledge should be organically combined to achieve the goal of recognizing both unknown objects and known dynamic objects with high accuracy.

## Figures and Tables

**Figure 1 biomimetics-07-00163-f001:**
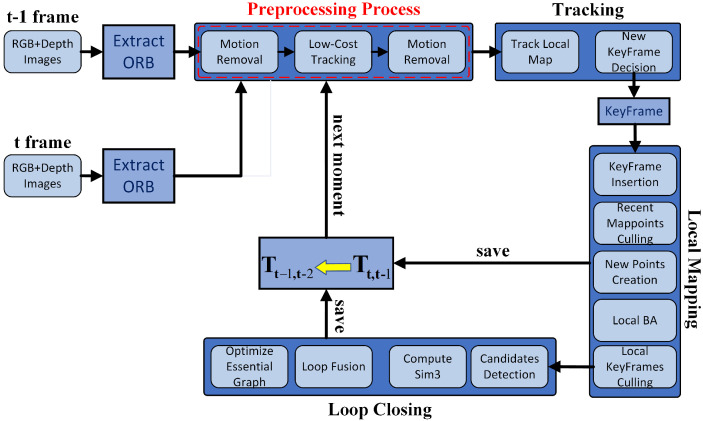
The DGFlow-SLAM frame proposed in this paper. The red dashed box means the preprocessing process part. The yellow arrow indicates that the optimized transformation matrix $T_{t,t-1}$ at the current moment is saved and used as $T_{t-1,t-2}$ at the next moment to speed up the preprocessing process.

**Figure 2 biomimetics-07-00163-f002:**
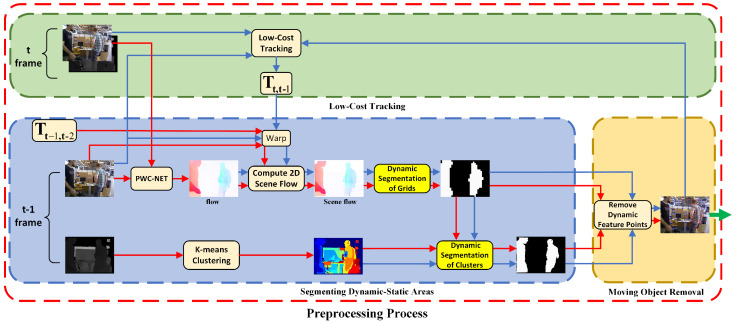
The red dashed line is an overview of the preprocessing process. Different color dashed lines indicate different modules. The blue module is used as the Segmenting Dynamic–Static Areas module to produce a grid segmentation dynamic–static mask image and a k-means clustering segmentation dynamic–static mask image. The yellow module is used as the Moving Object Removal module to remove dynamic keypoints using the two mask images jointly. The green module is used as a Low-Cost Tracking module to calculate rough poses. The blue and yellow modules together form the Motion Removal module in Figure 1, and the green module is the Low-Cost Tracking module in Figure 1. In this figure, we have two iterative processes, the red arrows indicate the first iterative process and the blue arrows indicate the second iterative process. The green arrow indicates the dynamic–static feature points obtained from the preprocessing process are passed to the tracking module. The highlighted parts are grid dynamic–static segmentation and k-means clustering dynamic–static segmentation, respectively.

**Figure 3 biomimetics-07-00163-f003:**
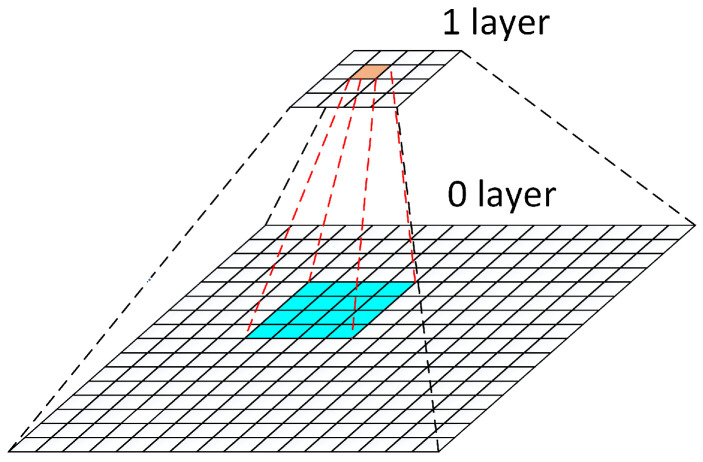
A two-level image pyramid to help speed up computing. The blue pixel block is the mapping result of the brown pixel block in layer 0.

**Figure 4 biomimetics-07-00163-f004:**
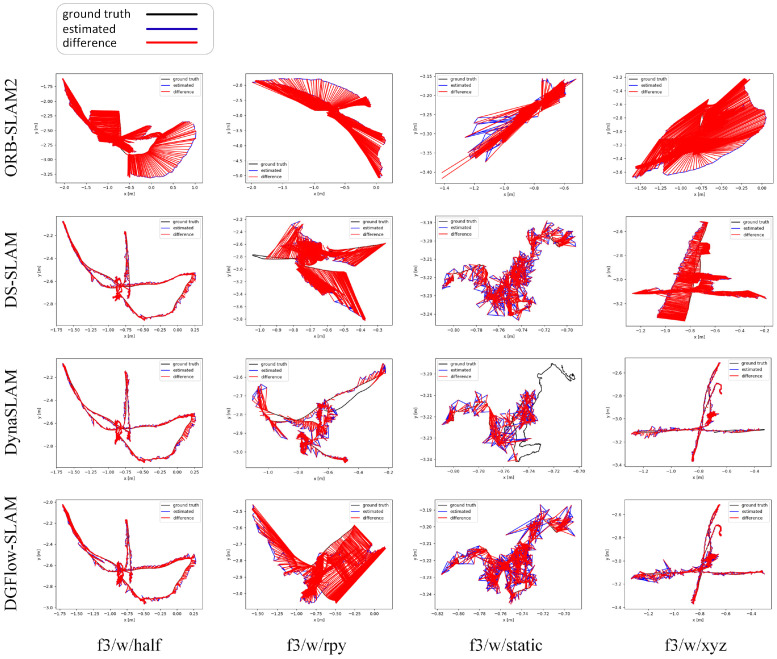
This is the camera trajectory image. Black lines are the true trajectory and blue lines are our estimated camera trajectory. Except for w/rpy, on the other three datasets, our estimated trajectory is basically around the real trajectory.

**Figure 5 biomimetics-07-00163-f005:**
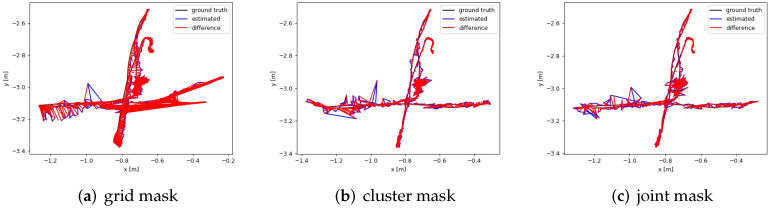
Camera trajectory image of the removal effect of three mask images on dynamic objects.

**Figure 6 biomimetics-07-00163-f006:**
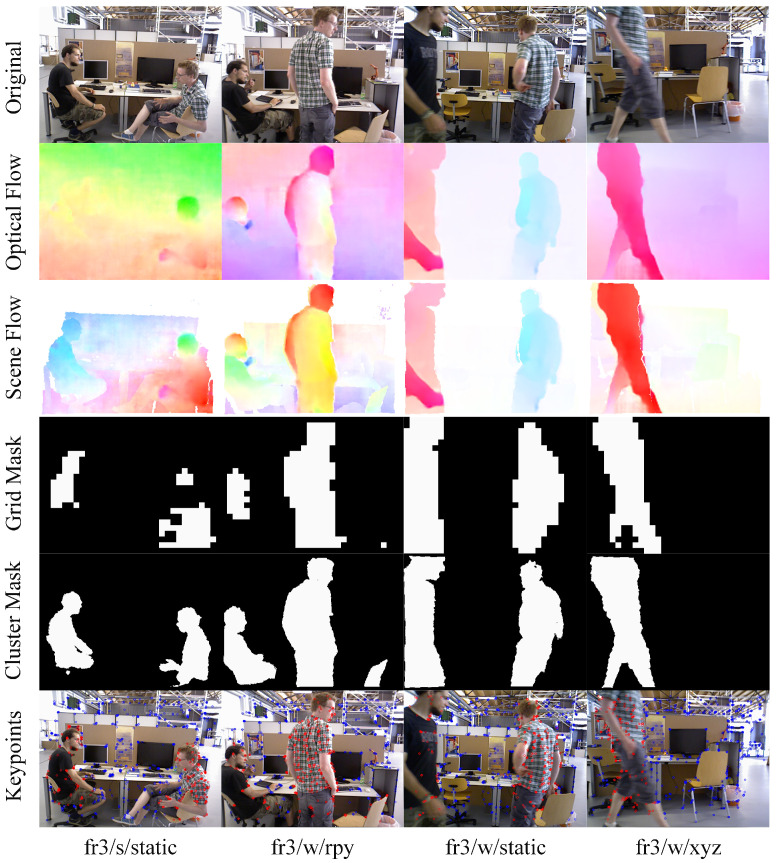
Comparison of the four datasets. The first row represents the input image, and the second row is the optical flow result produced by PWC-NET [39]. Our computed scene flow graph is shown in the third row. The fourth and the fifth rows are the dynamic–static grid mask images and the dynamic–static mean clustering mask images, respectively. The last row shows the distinction between the dynamic points and the static key points of ORB, and the red key points and the blue key points represent dynamic points and background points, respectively.

**Figure 7 biomimetics-07-00163-f007:**
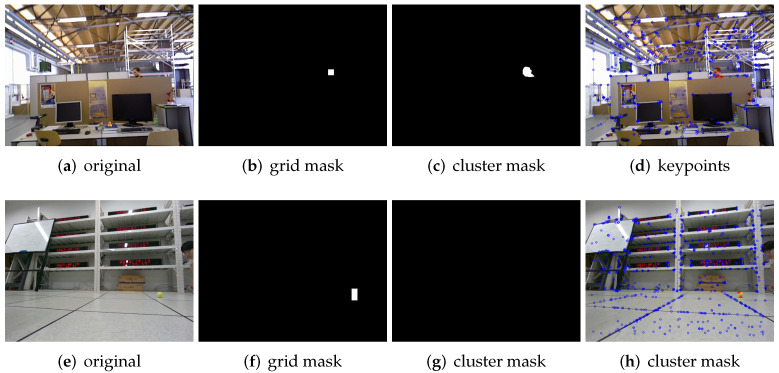
(**a**–**d**) show the cases of small object motion on the public data sets, as does Figure 5. (**e**,**f**) show the cases of small object motion taken with camera. In (**a**), only a head of the human body appears in the image. (**b**) shows the results using Algorithm 1. (**c**) shows the results using Algorithm 1. (**d**) shows the dynamical key points and the static key points using the algorithm DGFlow-SLAM. (**e**) shows an image of a small ball rolling, and (**f**) is the result of Algorithm 1. (**g**) is the result of Algorithm 2. (**h**) shows the dynamical key points and static key points using the algorithm DGFlow-SLAM.

**Figure 8 biomimetics-07-00163-f008:**
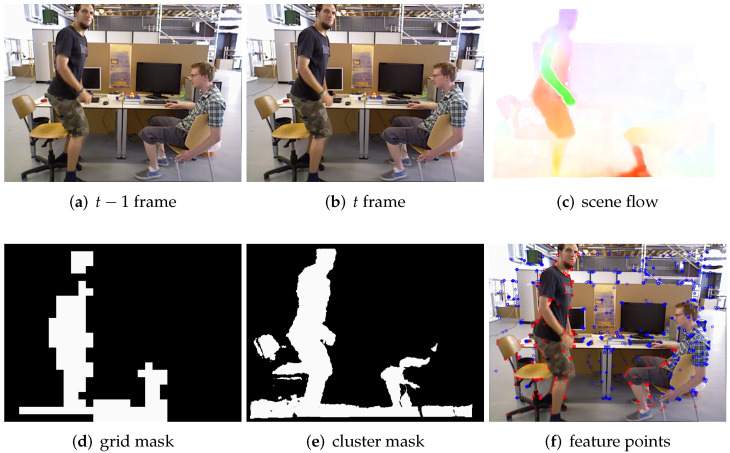
This figure shows an example of slight motion segmentation. (**a**,**b**) are frames at time t−1 and time *t*, respectively, in which the human body is in slow motion. (**c**) is a scene flow graph.

**Figure 9 biomimetics-07-00163-f009:**
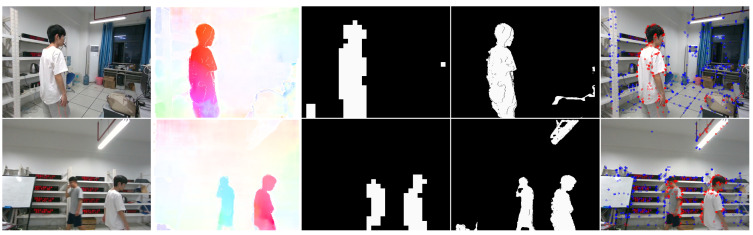
The experimental cases in our real environment. The first row is a scene of a single person exercising, and the second line shows two people walking in opposite directions. The white in the two mask images represents the dynamic part, and the black represents the background area.

**Table 1 biomimetics-07-00163-t001:** Result of absolute trajectory error (ATE).

Sequence	ORB-SLAM2		DS-SLAM		DynaSLAM		DGFlow-SLAM		Enhancements	
	RMSE	S.D.	RMSE	S.D.	RMSE	S.D.	RMSE	S.D.	RMSE	S.D.
w/half	0.5380	0.2374	0.0303	0.0159	**0.0296**	0.0157	0.0489	0.0284	90.91%	88.04%
w/rpy	1.0061	0.5496	0.4442	0.2650	**0.0354**	0.019	0.2789	0.1856	71.28%	66.28%
w/static	0.2036	0.0994	0.0081	0.0033	**0.0068**	0.0032	0.0086	0.0046	95.58%	95.37%
w/xyz	0.6445	0.2676	0.0247	0.0161	**0.0164**	0.0086	0.0280	0.0187	95.66%	93.01%
s/static	0.0080	0.0040	0.0065	0.0033	0.0108	0.0056	**0.0059**	**0.0029**	26.25%	27.5%

**Table 2 biomimetics-07-00163-t002:** Results of translation relative pose error (RPE translation).

Sequence	ORB-SLAM2		DS-SLAM		DynaSLAM		DGFlow-SLAM		Enhancements	
	RMSE	S.D.	RMSE	S.D.	RMSE	S.D.	RMSE	S.D.	RMSE	S.D.
w/half	0.3728	0.2873	0.0297	0.0152	**0.0284**	0.0149	0.0447	0.0284	88.01%	94.81%
w/rpy	0.4058	0.2964	0.1503	0.1168	**0.0448**	0.0262	0.1970	0.1492	51.54%	49.66%
w/static	0.1858	0.1614	0.0102	0.0038	**0.0089**	0.0044	0.0114	0.0062	93.86%	96.16%
w/xyz	0.4214	0.2923	0.0333	0.0229	**0.0217**	0.0119	0.0398	0.0266	90.56%	90.90%
s/static	0.0084	0.0041	0.0078	0.0038	0.0126	0.0067	**0.0076**	**0.0037**	9.52%	9.76%

**Table 3 biomimetics-07-00163-t003:** Results of rotation relative pose error (RPE rotation).

Sequence	ORB-SLAM2		DS-SLAM		DynaSLAM		DGFlow-SLAM		Enhancements	
	RMSE	S.D.	RMSE	S.D.	RMSE	S.D.	RMSE	S.D.	RMSE	S.D.
w/half	7.6591	5.8202	0.8142	0.4101	0.7842	0.4012	1.0696	0.6817	86.03%	88.29%
w/rpy	7.8952	5.7114	3.0042	2.3065	0.9894	0.5701	3.7283	2.7804	52.18%	51.32%
w/static	3.4029	2.9116	0.2690	0.1215	0.2612	0.1259	0.2800	0.1307	91.54%	95.11%
w/xyz	8.0446	5.5260	0.8266	0.2626	0.6284	0.3848	**0.6187**	0.3754	92.31%	93.21%
s/static	0.2729	0.1179	0.2735	0.1215	0.3416	0.1642	**0.2698**	**0.1150**	**1.14%**	**2.46%**

**Table 4 biomimetics-07-00163-t004:** The time cost of each module of our system.

Sequence	ORB Extraction and Matching	Removal of Outliers	PWC-NET	Total Time
Times (ms)	21	18	12	51

**Table 5 biomimetics-07-00163-t005:** Absolute trajectory error obtained by the DGFlow-SLAM system after removing dynamic points using three mask images separately.

Mask Image	Grid Mask	Kmean Mask	Joint Mask
RMSE	0.1259	0.0275	0.0234

## Data Availability

Not applicable.

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
