# Peer review of "DGFlow-SLAM: A Novel Dynamic Environment RGB-D SLAM without Prior Semantic Knowledge Based on Grid Segmentation of Scene Flow"

_biomimetics, 2022, doi:10.3390/biomimetics7040163_

Round 1

Reviewer 1 Report

In this paper, the authors propose a dynamic environment RGB-D SLAM without prior knowledge based on grid segmentation , but there are still some problems.   Please consider the following comments: 1. How to define no prior knowledge? Since you use PWC-NET neural network to estimate the estimate the optical flow, I think prior knowledge is applied. 2. While you combine the segmenting grid mask image and the segmenting mean clustering mask image for processing, why the running speed of your method is faster than DynaSLAM and DS-SLAM. Do you perform all the experiments on the same platform? 3. I think an ablation study(experiment) is needed to verify the effectiveness of “Dynamic Grid Segmentation” and “Segmentation Algorithm Based on Deep k-means Clustering”, and show the combination of both can achieve better performance. 4. There are some mistakes in the writing of the paper, such as “where t0 and mean the initial time and the transform matrix from” in page 5. 5. The authors should carefully check the format of references. They should be consistent(2 has pp before page number while 1 does not have). 6. The theoretical contribution of the work should be clarified when compared with existing methods. Also, it is necessary to theoretically analyze why the proposed method is better than existing methods.

Reviewer 2 Report

This paper propose a new framework to reduce the impact of dynamic feature points for the ORB-SLAM2. The framework uses the grid segmentation and the k-mean clustering segmentation to remove the dynamic regions. The experiments show that the violent motion scene and slight motion scene can be detected. The following are opinions:

What is the standard for distinguishing the violent motion scene and slight motion scene, is it distinguished by the speed of the moving object? The speed of an object is less than how many meters per second is the slight motion scene? What is the minimum moving speed of the object that the algorithm in this paper can distinguish?

The paper aimed at "To enhance the adaptability of the classic ORB-SLAM2...", then, whether the algorithm can distinguish the environment in which there are dynamic objects near and far in the outdoor dataset, such as the street dataset in the kitti dataset. It is better to explain in detail in the text.

In line 423, "the classical ORB-SLAM", is ORB-SLAM2.

Reviewer 3 Report

The following is the report on this manuscript.

The subject is interesting, but at the moment the manuscript is extremely broad with respect to the proposed method for replicability and lacks clarity in some parts.

The authors of this manuscript built on previous work that used the semantic SLAM systems incorporating a grid segmentation method and an adaptative threshold method in enhancing dynamic object detection. The relevance of this work is not well addressed, partly because the author’s motivation/justification does not address the real benefit for users in terms of using the DGFlow-SLAM instead of ORB-SLAM, DynaSLAM and DS/SLAM. This is because there is no clear workflow in which the method can be replicable and there is not enough statistical evidence about the accuracy, and fasten the time of calculation in different processes of dynamic object identification with respect to the previous SLAM systems. Although calculation obeys a dynamic between scenes. The science sounds in the manuscripts, however, in the current state the manuscripts need major revisions to be publishable.

Please find below the detailed comments.

First, there are some lines without numbering. Between lines 220 -221 there are 13 lines without numbers. Between lines 258 – 259 there are 7 lines without numbers. Then between lines 271 and 272, there are 25 without a line number. Between lines 275 - 276, there are 10 lines without numbers. Between Lines 37 – 374 there are 6 lines without numbers. Between lines 411 – 412, in Figure 7 there is no line numbering.

When you mention the potential dynamic targets (line 10), there is no reference in the text that show the assortments of these targets. This will give an interesting point to test the DGFlow-SLAM framework. If the experiment runs over dataset TUM associated with the potential dynamic targets. You have to clarify this from the beginning.

Lines 35 – 63. I’ll suggest splitting this paragraph. This is too long and difficult for the comprehension of the ideas exposed. At least split from lines 35-52 and then 52 – 63.

Line 61 -63: some references are needed here.

 Line 94: Please check if all references correspond to SLAM systems. For example, reference 22, about Korhonen’s Self-Organizing Maps is not a SLAM system itself. The method can be used for codebook vector segmentation but is not a SLAM system.

Line 100 – 141. This is a 42 lines paragraph. Please check, since the central idea behind this paragraph is obscure and readers easy lost the comprehensive flow of the manuscript. I’ll suggest reducing or including the SLAM background in a table highlighting the central advances in SLAM systems.

Line 143: Change “The researchers” by Previous studies.

Figure 1: If a novel framework/methodology is presented in this manuscript, please provide a complete workflow to ensure replicability for Biomimetics readers. This figure how is presented summarizes the method but does not ensure others researchers replicate and use the DGFlow-SLAM framework. Besides, you need to highlight in this figure the part in which grid segmentation and the adaptative threshold method contribute to the improved DGFlow-SLAM. From here is needed to understand the benefit of the framework with respect to previous SLAM systems.

Also, a comprehensive workflow that corresponds to section 3 headers are needed. If you describe it in section 3.2 as preprocessing process, this is not clear or indicated in Figure 1.

In Figure 2, you detailed the preprocessing process but the font is too small for reading inside each box. In the footnote, there is no explanation about green, blue and yellow boxes even for red or black dashed lines, and for blue and red arrows. I know that this is in the text but the footnote of the figure is also needed.

Note that some of the equations are not explained explicitly with Equation numbering in the text. Please check.

Line 228: a reference for 7 meters is needed here.

Line 237: 22ms is a result? If not please add a reference, if yes move this to the results section.

Line 243: A formal explanation/reference about the N grid regions of 20x20 is needed. If this is your own development please explain why you decide this or if a process to reach this was done.

Line 252: Clarify what is a valid or invalid grid before explaining the dynamic-static segmentation method.

You only describe Equation 5 with references in the text, but you describe 11 equations without references. Please if this is your own development add equation references across the text explanation If these are equations adapted from previous studies please add the references.

Equation 6: Note that exponents in equation 6 are not the same that you describe in line 267. The reader can confuse between this and optical flow vectors defined in line after equation 8. Please check.

Line 272: Add a reference for 1500~2000.

Line 275: I select this row but the comment is for 3 lines below this row in which you describe the 24 blocks. This number needs a deep/formal explanation about this number. When you applied the k-means how did you select the number of classes to segment the image? Do you use the elbow rule to set this?

Did you set up 24 blocks for all experiments? This is constant if algorithm 2. This needs a critical view when discussing the results.

Line 301: ATE and RPE need references.

Line 316: You need a metric to justify this about ORB-SLAM. Not only from visual inspection. Even if this is true you need a formal metric to argue this performance of the method. Or mention that this is quantitatively demonstrated in section 4.2.2

Figure 3. The caption legend for black, red and blue lines is unreadable in the figure. Please increase font sizes or if it is a common legend please add just one legend for all cases.

Please add a row description for all captions in Figure 4.

Equation 13. If this is a quantitative metric for asses the method performance. This will be described in the methods section and not here which is intended to present results and the discussion.

The discussion of the manuscript is scarce there is no comparison and contrast of the framework exposed. You need to improve the discussion of your results. The current state of the discussion does not reflect the improvements in the framework presented. You have to centre all your efforts on improving this section. The only comparative aspect addressing comparison with other works is the runtime evaluation in lines 389 – 401. You have to present a similar discussion for the other aspects that advantage de DGFlow-SLAM with respect to other methods.  Some criticism of your framework is also needed to give hints for future improvement.

Figure 7. In the footnote, there is a double “the The”

The conclusion section needs to be reformulated to show the DGFlow-SLAM improvements with clarity and highlight the aspect that your improvements respect other SLAM systems approaches. The conclusion section how is presented summarized the manuscript but does not conclude about the suitability and advantages of using this SLAM framework. Just the last two phrases present the conclusion. Please, use this approach to include all relevant/important aspects of your research that improve the actual state of SLAM systems. Besides, left open the door/ hints for future developments for improving SLAM systems even the DGFlow-SLAM framework.

Round 2

Reviewer 1 Report

I have no comments. My concerns have been addressed.

Reviewer 3 Report

All my queries were solved satisfactorily.